# Pruning Cannot Hurt Robustness: Certified Trade-offs in Reinforcement Learning

## Abstract

Reinforcement learning (RL) policies deployed in real-world environments must remain reliable under adversarial perturbations. At the same time, modern deep RL agents are heavily overparameterized, raising costs and fragility concerns. While pruning has been shown to improve robustness in supervised learning, its role in adversarial RL remains poorly understood. We develop the first theoretical framework for *certified robustness under pruning* in state-adversarial Markov decision processes (SA-MDPs). For Gaussian and categorical policies with Lipschitz networks, we prove that elementwise pruning can only tighten certified robustness bounds; pruning never makes the policy less robust. Building on this, we derive a novel three-term regret decomposition that disentangles clean-task performance, pruning-induced performance loss, and robustness gains, exposing a fundamental performance–robustness frontier. Empirically, we evaluate magnitude and micro-pruning schedules on continuous-control benchmarks with strong policy-aware adversaries. Across tasks, pruning consistently uncovers reproducible "sweet spots" at moderate sparsity levels, where robustness improves substantially without harming—and sometimes even enhancing—clean performance. These results position pruning not merely as a compression tool but as a structural intervention for robust RL.

## 1 Introduction

Reinforcement learning (RL) has demonstrated impressive capabilities in domains ranging from strategic games (Silver et al., 2017) to robotic control (Lillicrap et al., 2016). RL is now employed in various safety-critical applications, such as for autonomous vehicles (Kendall et al., 2019), computer network defence (Foley et al., 2022), and language model alignment (Ouyang et al., 2022), often without human-in-the-loop supervision. It is therefore of crucial importance to consider how robust RL policies are against malicious actors who would seek to adversarially manipulate their actions, and how we might better defend against such attacks.

Modern model-free RL policies are typically over-parameterized (Sokar et al., 2023; Thomas, 2022), which makes them more expensive to deploy and fragile to distribution shift (Kumar et al., 2022; Menon et al., 2021). A natural solution is pruning, which has been widely explored in supervised learning for model compression (Hayou et al., 2021), improved generalization (LeCun et al., 1989), and robustness to adversarial attacks (Sehwag et al., 2020; Li et al., 2023). However, unlike in supervised learning, the relationship between pruning and robustness in RL remains largely unexplored (Graesser et al., 2022; Zhang et al., 2020). Reinforcement learning poses unique challenges: perturbations to observations can propagate and accumulate over long-horizon trajectories, where even small errors may compound into catastrophic failures (Weng et al., 2020).

In this work, we study *sparse RL robustness* to examine how pruning influences both benign performance and adversarial robustness, and how these often competing objectives can be better aligned. We model adversaries which perturb agent observations through a state adversarial Markov decision process (Figure 1), building off work from Zhang et al. (2020). This can be used to show that pruning offers theoretical guarantees, proving that **element-wise pruning cannot worsen the bounds of certified robustness**. We derive a three-term regret decomposition that disentangles clean performance, pruning-induced performance loss, and robustness gain.

We validate these predictions experimentally using Proximal Policy Optimisation (PPO) (Schulman et al., 2017) across multiple continuous control environments under a range of strong policy-aware adversaries. Across settings, pruning consistently uncovers a "sweet spot" at intermediate sparsities where robustness improves substantially without sacrificing — and sometimes even enhancing — clean performance. We combine pruning with state-adversarial regularisation (Zhang et al., 2020) to highlight its effectiveness as a complementary technique to existing robustness measures. Across three MuJoCo benchmarks, pruning achieves up to 25% higher certified robustness while maintaining at least 95% of baseline clean performance, consistently revealing reproducible Pareto optima.

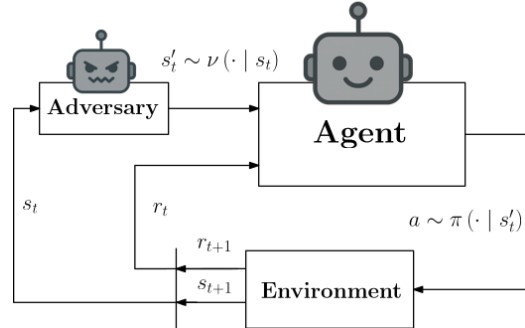

Figure 1: The SA-MDP framework. A victim agent receives a perturbed observation from an adversary trained to reduce its performance. The true state $s_t$ is emitted by the environment, tampered by the adversary, and then passed to the victim.

**Our contributions.**

*Theoretical guarantees:* We prove that pruning monotonically improves certified robustness in SA-MDPs, establishing that sparsity cannot reduce adversarial resilience.

*Trade-off characterization:* We derive a three-term regret bound that formalizes the interplay between pruning, clean performance, and robustness, clarifying when these align or conflict.

*Empirical validation:* We show, across continuous-control benchmarks, that pruning consistently uncovers reproducible "sweet spots" where robustness gains outweigh performance losses.

## 2 RELATED WORK

### 2.1 ADVERSARIAL ATTACKS AND ROBUSTNESS IN RL

**Training-phase attacks.** A first class of training-phase attacks are reward attacks. As rewards formally characterize an agent's purpose, altering the rewards logically changes the learned policies of the agents. A reward-poisoning attack was proposed in batch RL (Zhang & Parkes, 2008; Zhang et al., 2009), where rewards were stored in an unlocked, pre-collected dataset. This provided the attacker with the opportunity to directly change the reward in the dataset. Variants of this attack have also been proposed (Huang & Zhu, 2019; Rakhsha et al., 2020; 2021; Cai et al., 2022). These variants use different oracle access to the model being attacked. It is further possible to attack an RL agent, without tampering the reward. For example, Xu et al. (2021); Xu (2022) propose an environment-poisoning attack, where the victim RL agent is misled by subtle changes to the environment. RL agents can also be attacked by embedding triggers that elicit malicious behaviour (Kiourti et al., 2020; Yu et al., 2022). Here, the attacker alters the training process so that the agent learns to associate a rare pattern (the trigger) with an attacker-chosen behaviour. We remark that training-phase attacks have also been studied for RL from Human Feedback (RLHF), mostly in the context of LLMs fine-tuning, e.g., (Wang et al., 2024; Chen et al., 2024; Shi et al., 2023; Rando & Tramèr, 2024; Zhao et al., 2024).

**Test-phase attacks.** Test-phase attacks aim to deceive a trained policy. One common approach involves introducing perturbations into the state space at different points during execution (Huang et al., 2017; Lin et al., 2017; Kos & Song, 2017). Beyond this, carefully crafted perturbation sequences can be designed to steer agents toward specific states Behzadan & Munir (2017); Lin et al. (2017); Tretschk et al. (2018); Weng et al. (2020); Hussenot et al. (2020); Mo et al. (2023). Such perturbation-based attacks, however, can be mitigated using techniques that reinforce cumulative rewards Chan et al. (2020). In addition, research has explored test-phase transferability attacks Huang et al. (2017); Yang et al. (2020); Inkawhich et al. (2020), which exploit the empirical observation that adversarial examples crafted to deceive one model (a surrogate) can also mislead other models, even when those models differ in architecture, training data, or parameters. More recently, test-phase attacks have also been devised to specifically target RLHF in LLMs (Wang et al., 2023; Xi

et al., 2025; Liu et al., 2024; Zhu et al., 2023). For further information we refer interested readers to surveys such as Das et al. (2025); Yao et al. (2023); Shayegani et al. (2023).

**Robustness in RL.** While robustness in RL has been extensively explored, studies specifically addressing adversarial robustness remain limited. Empirical robust learning typically uses heuristics or evaluations to enhance model reliability. An effective way of improving robustness is to use Adversarially Robust Policy Learning (ARPL), which incorporates physically plausible adversarial examples during training (Mandlekar et al., 2017; Tessler et al., 2019; Zhou et al., 2024). It is likewise possible to make agents more resilient, by altering the environment during training Jiang et al. (2021); Parker-Holder et al. (2022); Dennis et al. (2020). Additional contributions include Ball et al. (2021), showing that Augmented World Models improve generalization and Ball et al. (2020), where agents use a context variable to adapt to changes in environment dynamics.

## 2.2 PRUNING

Sparsity is valuable not only for model compression and faster inference (Han et al., 2015; Molchanov et al., 2017), but also for improving generalisation (LeCun et al., 1989; Hassibi & Stork, 1992; Bartoldson et al., 2020). Pruning design choices include whether to remove individual parameters (LeCun et al., 1989; Han et al., 2015) or use structured sparsity (Wen et al., 2016; Lasby et al., 2024), and whether to prune statically (Frankle & Carbin, 2019) or dynamically during training (Evci et al., 2020; Mocanu et al., 2018; Prechelt, 1997). Criteria include random selection (Liu et al., 2022), magnitude (LeCun et al., 1989), saliency (Hassibi & Stork, 1992), or evolutionary strategies (Mocanu et al., 2018), often paired with Straight-Through Estimators (Vanderschueren & Vleeschouwer, 2023; Bengio et al., 2013; Hinton, 2012) for gradient flow through binary masks. For a detailed overview, see, e.g., Cheng et al. (2024).

For supervised learning, it has been empirically demonstrated that pruning can improve robustness against adversarial attacks, both through the above methods and augmenting with additional adversarial objectives. In Cosentino et al. (2019) lottery tickets (Frankle & Carbin, 2019), pruned up to $\sim 96\%$, can outperform the original network on adversarial accuracy. Work in Fu et al. (2021) extends Frankle & Carbin (2019); Malach et al. (2020) to show that tickets exist which can outperform the dense network on adversarial examples, without any training. HYRDA (Sehwag et al., 2020) creates a risk minimisation objective for pruning which optimises the pruning to be adversarially robust. Conversely, Cosentino et al. (2019) separately applies pruning followed by adversarial training (Madry et al., 2018) to produce more robust sparse networks and Bair et al. (2024) introduces a sharpness-aware pruning criterion to encourage flatter, more generalisable networks. In contrast, far less work has been done to understand the interaction of sparsity and robustness for RL policies, motivating this work.

## 3 ROBUSTNESS BOUNDS IN SA-MDPS

### 3.1 SETTING

We study a state-adversarial Markov decision process (SA-MDP) defined with perturbation sets $B(s) \subseteq \mathcal{S}$. A standard MDP is specified as a tuple $(\mathcal{S}, \mathcal{A}, R, p, \gamma)$, where a stationary stochastic policy is given by $\pi_\theta : \mathcal{S} \to \mathcal{P}(\mathcal{A})$ with density $\pi_\theta(a|s)$. In the SA-MDP setting, the agent does not act on the true state $s$ but instead observes an adversarially perturbed state $\nu(s) \in B(s)$ and selects actions according to $\pi_\theta(\cdot|\nu(s))$, while the environment transitions based on the true state through $p(\cdot|s, a)$. Consequently, an SA-MDP can be represented as $(\mathcal{S}, \mathcal{A}, B, R, p, \gamma)$. in this work, we constrain $\nu$ to an $\ell_p$ ball: $B(s) := \{\hat{s} \in \mathcal{S} : \|\hat{s} - s\|_p \le \varepsilon\}$ with budget $\varepsilon > 0$ and $p \in \{2, \infty\}$.

For distributions $P, Q$ on $\mathcal{A}$, we define the total variation distance as

$$D_{\mathrm{TV}}(P, Q) := \sup_{E \subseteq \mathcal{A}} |P(E) - Q(E)|.$$

For each state $s \in \mathcal{S}$, we define $TV_{\max}(s; \theta) := \max_{\hat{s} \in B(s)} D_{\mathrm{TV}}(\pi_\theta(\cdot|s), \pi_\theta(\cdot|\hat{s}))$. Let $d_\mu^{\pi_\theta}$ be the discounted visitation distribution from $\mu$, and set

$$F(\theta) := \mathbb{E}_{s \sim d_\mu^{\pi_\theta}}[TV_{\max}(s; \theta)], \qquad \mathcal{B}(\theta) := \alpha F(\theta) \text{ with } \alpha = 2\Big[1 + \tfrac{\gamma}{(1-\gamma)^2}\Big] R_{\max},$$

with $|R(s, a, s')| \leq R_{\max}$.

**Policy classes and constants.** We consider (i) Gaussian policies $\pi_\theta(a|s) = \mathcal{N}(\mu_\theta(s), \Sigma)$ with fixed $\Sigma \succ 0$, and (ii) categorical policies $\pi_\theta(\cdot|s) = \mathrm{softmax}(z_\theta(s))$, $z_\theta(s) \in \mathbb{R}^K$. In subsequent bounds we use the constant $c = (\sqrt{2\pi \lambda_{\min}(\Sigma)})^{-1}$ for Gaussian policies and $c = 1/4$ for categorical softmax policies. We write $\tilde{V}^{\pi_\theta \circ \nu^*}$ for the robust value under the optimal adversary. The robustness gap for a state $s \in \mathcal{S}$ is $V^{\pi_{\theta'}}(s) - \tilde{V}^{\pi_{\theta'} \circ \nu^*(\pi_{\theta'})}(s)$.

**Additional notation.** We use $\|x\|_p$ for vector $\ell_p$ norms; $\|W\|_F$ (Frobenius), $\|W\|_1 = \max_j \sum_i |W_{ij}|$, $\|W\|_\infty = \max_i \sum_j |W_{ij}|$ for matrices; and $\|J\|_{\mathrm{op}}$ for spectral norm. For neural policies, $g_\theta(s)$ denotes logits/means and $J_{g_\theta}(s)$ its Jacobian.

## 3.2 PERFORMANCE–ROBUSTNESS TRADE-OFFS

We show that elementwise pruning of a policy network in stochastic action MDPs cannot worsen its certified robustness guarantee. This result follows from a surrogate Lipschitz bound, which decreases under pruning, thereby ensuring monotone improvement in robustness.

**Theorem 1** (SA-MDP robustness improves under pruning). *Let $\pi_\theta$ be either a Gaussian or categorical policy realized by a feedforward network with Lipschitz activations $\sigma_\ell$ and weights $\theta$. Define the surrogate Lipschitz bound*

$$\tilde{L}_\theta := \left( \prod_{\ell=1}^{L-1} L_{\sigma_\ell} \right) \prod_{\ell=1}^{L} \min\left\{ \|W_\ell\|_F, \ \sqrt{\|W_\ell\|_1 \|W_\ell\|_\infty} \right\}.$$

*Let $\theta'$ be obtained from $\theta$ by elementwise pruning. Then*

$$\max_s \{ V^{\pi_\theta}(s) - \tilde{V}^{\pi_\theta \circ \nu^*(\pi_\theta)}(s) \} \ \leq \ \alpha \, c \, \tilde{L}_\theta \, \varepsilon,$$

*and*

$$\max_s \{ V^{\pi_{\theta'}}(s) - \tilde{V}^{\pi_{\theta'} \circ \nu^*(\pi_{\theta'})}(s) \} \ \leq \ \alpha \, c \, \tilde{L}_{\theta'} \, \varepsilon \ \leq \ \alpha \, c \, \tilde{L}_\theta \, \varepsilon,$$

*Thus, under pure elementwise pruning, the certified robustness bound is monotone nonincreasing.*

Intuitively, this theorem shows that pruning reduces the network's sensitivity to perturbations, so the certified robustness of the policy can only improve as parameters are removed.

**Training remark.** The monotonicity result (Theorem 1) applies to pruning on a fixed set of weights. During training, gradient steps may enlarge weight norms and hence $\tilde{L}_\theta$, so robustness is not globally monotone. Nevertheless, each pruning step strictly decreases $\tilde{L}_\theta$ relative to the current parameters, acting as a monotone regularizer counteracting weight growth. This explains why robustness tends to improve steadily in practice (Sec. 5) when pruning is interleaved with training.

**Tightness of the bound.** While Theorem 1 provides a provably monotone global robustness bound, it can be loose in practice. A sharper, distribution–dependent refinement is given in Lemma 1 (Appendix), which often yields much tighter estimates, though without the same monotonicity guarantee under pruning.

Theorem 1 guarantees pruning cannot worsen the worst-case robustness gap. However, worst-case bounds can be overly pessimistic. To obtain guarantees that better capture typical performance, we next consider expected versions of the robustness gap, aligned with the population-level objective $F(\theta)$. This motivates our second main result, which characterizes robustness through $F(\theta)$ and bounds the expected degradation in value under the optimal adversary.

**Theorem 2** (Unified regret under SA attack). *Fix a start distribution $\mu$. Write $J(\pi) := \mathbb{E}_{s_0 \sim \mu}[V^\pi(s_0)]$ and $\tilde{J}(\pi) := \mathbb{E}_{s_0 \sim \mu}[\tilde{V}^\pi(s_0)]$. Let $\bar{\pi}$ be any comparator policy (e.g., $\pi^*$ maximizing $J$ or $\tilde{\pi}^*$ maximizing $\tilde{J}$), and let $\nu^*$ denote the optimal SA adversary for $\pi_\theta$. Define*

$$\mathrm{Reg}_{\mathrm{clean}}(\theta; \bar{\pi}) := J(\bar{\pi}) - J(\pi_\theta), \qquad \mathrm{Reg}_{\mathrm{atk}}(\theta; \bar{\pi}) := J(\bar{\pi}) - \tilde{J}(\pi_\theta \circ \nu^*).$$

*Then, it holds*

$$\mathrm{Reg}_{\mathrm{atk}}(\theta; \bar{\pi}) - \mathrm{Reg}_{\mathrm{clean}}(\theta; \bar{\pi}) = J(\pi_\theta) - \tilde{J}(\pi_\theta \circ \nu^*) \ \leq \ \mathcal{B}(\theta).$$

*Additionally, if $\pi_\theta$ is Gaussian with fixed $\Sigma \succ 0$ or categorical softmax with Lipschitz network, then*

$$\mathrm{Reg}_{\mathrm{atk}}(\theta; \bar{\pi}) - \mathrm{Reg}_{\mathrm{clean}}(\theta; \bar{\pi}) \ \leq \ \alpha\, c\, \tilde{L}_\theta\, \varepsilon.$$

*Moreover, if $\theta'$ is obtained by entrywise pruning, then*

$$\mathrm{Reg}_{\mathrm{atk}}(\theta'; \bar{\pi}) - \mathrm{Reg}_{\mathrm{clean}}(\theta'; \bar{\pi}) \ \leq \ \alpha\, c\, \tilde{L}_{\theta'}\, \varepsilon \ \leq \ \alpha\, c\, \tilde{L}_\theta\, \varepsilon.$$

Theorem 2 shows that the extra regret a policy suffers under the optimal state-adversarial attack (compared to its clean regret) is always bounded by a robustness coefficient $\mathcal{B}(\theta)$, and in particular by the Lipschitz surrogate $\tilde{L}_\theta$ for Gaussian or softmax policies. In other words, pruning cannot increase this attack–clean regret gap and in fact makes the bound tighter.

**Pruning sensitivity.** For pruned parameters $\theta' = \theta - \Delta\theta$, define the path–averaged parameter sensitivity

$$\mathcal{L}_{\mathrm{par}}(\theta, \theta') := \int_0^1 \left( \mathbb{E}_{s \sim d_\mu^{\pi_\theta}} \| J_\phi g_\phi(s) \|_{\mathrm{op}}^2 \right)_{\phi = \theta' + t(\theta - \theta')}^{1/2} \mathrm{d}t,$$

with $g_\phi = \mu_\phi$, for Gaussian policies and $g_\phi = z_\phi$ for categorical. Lemma 3 (Appendix) shows that both clean and attacked value drops satisfy

$$J(\pi_\theta) - J(\pi_{\theta'}), \ \ \tilde{J}(\pi_\theta \circ \nu^*) - \tilde{J}(\pi_{\theta'} \circ \nu^*) \ \leq \ \alpha c\, \mathcal{L}_{\mathrm{par}}(\theta, \theta')\, \|\Delta\theta\|.$$

This term serves as the performance loss from pruning in Theorem 3, complementing the baseline regret and robustness gap to yield the full three–term trade-off.

**Theorem 3** (Performance–robustness trade-off under pruning). *Fix any comparator policy $\bar{\pi}$. For pruned parameters $\theta' = \theta - \Delta\theta$,*

$$\mathrm{Reg}_{\mathrm{atk}}(\theta'; \bar{\pi}) \ \leq \ \underbrace{\mathrm{Reg}_{\mathrm{clean}}(\theta; \bar{\pi})}_{\text{clean regret of unpruned}} + \underbrace{\alpha\, c\, \mathcal{L}_{\mathrm{par}}(\theta, \theta')\, \|\Delta\theta\|}_{\text{performance loss from pruning}} + \underbrace{\alpha\, c\, \tilde{L}_{\theta'}\, \varepsilon}_{\text{robustness gap of pruned}} \ .$$

**Interpretation.** The three–term bound exposes a fundamental performance–robustness trade–off. The first term is the clean regret of the unpruned policy, determined by baseline training quality. The second term, $\alpha c\, \mathcal{L}_{\mathrm{par}}(\theta, \theta') \|\Delta\theta\|$, is the performance loss from pruning. Here $\mathcal{L}_{\mathrm{par}}$ is a path–averaged sensitivity: it measures how strongly the policy's outputs react to parameter perturbations along the path from $\theta$ to $\theta'$. Low sensitivity implies pruning has little effect, while high sensitivity makes small weight changes costly. This explains why magnitude pruning is effective: removing small–magnitude weights keeps $\|\Delta\theta\|$ small, reducing the penalty.

The third term, $\alpha c\, \tilde{L}_{\theta'} \varepsilon$, is the robustness gap of the pruned policy, controlled by its input Lipschitz constant. Because pruning reduces $\tilde{L}_\theta$, this term always improves. Thus pruning simultaneously hurts via performance loss and helps via robustness, and the optimal sparsity balances these opposing effects. Pruning therefore acts not just as compression but as a structural intervention trading margin for robustness.

## 4 EXPERIMENTS AND RESULTS

We study the performance–robustness trade-off predicted by our SA-MDP theory under structured network sparsification. Concretely, we couple on-the-fly weight pruning with adversarially robust policy optimization on continuous-control benchmarks. This section specifies environments, policies, attacks, pruning strategies, and the full training–evaluation protocol.

### 4.1 TASKS AND POLICIES

We evaluate on three MuJoCo continuous-control tasks from Gym: `Hopper`, `Walker2d`, and `HalfCheetah`. Policies are stochastic Gaussian actors $\pi_\theta(a \mid s) = \mathcal{N}(\mu_\theta(s), \Sigma)$ with state-independent diagonal covariance $\Sigma$. Value functions use separate MLPs. Unless otherwise noted, both actor and critic are multilayer perceptrons (MLPs) with Lipschitz activations and standard initialization (full architecture details are provided in Appendix D).

Table 1: Summary of adversarial attacks used during training and evaluation.

| Attack | Description |
|---|---|
| Random | Samples $\hat{s}$ uniformly from the perturbation set $B(s)$. |
| Value-guided | Perturbs states to minimize $V^\pi(s)$ via gradient descent on the critic. |
| MAD | Maximizes $D_{\mathrm{KL}}(\pi_\theta(\cdot \mid s) \,\|\, \pi_\theta(\cdot \mid \hat{s}))$ with projected gradient steps. |
| Robust Sarsa (RS) | Uses a robust TD update of $Q^\pi$ to find perturbations $\hat{s}$ that minimize $Q^\pi(s, \pi(\hat{s}))$. |

## 4.2 State-Adversarial Training Objective

Our theory (Sec. 3) shows that robustness bounds are governed by divergences between $\pi_\theta(\cdot \mid s)$ and $\pi_\theta(\cdot \mid \hat{s})$ for perturbed states $\hat{s} \in B(s)$. By Pinsker's inequality, these total variation terms can be controlled by KL divergences. To operationalize this, we adopt the SA-regularization term introduced in prior work on robust PPO (Zhang et al. (2020)):

$$\mathcal{R}_{\mathrm{SA}}(\theta) \;=\; \mathbb{E}_{s \sim d_\mu^{\pi_\theta}}\Big[ \max_{\hat{s} \in B(s)} D_{KL}\big(\pi_\theta(\cdot \mid s) \,\|\, \pi_\theta(\cdot \mid \hat{s})\big)\Big],$$

where $D_{KL}$ is instantiated as KL divergence. While KL does not appear directly in the robustness bounds, it serves as a theoretically justified surrogate via Pinsker's inequality, and has been widely used in the literature on adversarially robust RL.

The actor objective becomes $\mathcal{L}_\pi(\theta) \;=\; \mathcal{L}_{\mathrm{PPO}}(\theta) \;+\; \kappa\, \mathcal{R}_{\mathrm{SA}}(\theta)$, where $\kappa \geq 0$ toggles the regularization strength. Setting $\kappa = 0$ disables SA regularization, yielding pruning-only training.

**Perturbation sets.** For state attacks we use $\ell_\infty$ balls $B(s) = \{\hat{s} : \|\hat{s} - s\|_\infty \leq \varepsilon\}$ in normalized state space, matching the SA-MDP formulation and robust PPO practice, with environment–specific budgets $\varepsilon = 0.075$ (`hopper`), $0.05$ (`walker2d`), $0.15$ (`ant`), and $0.15$ (`halfcheetah`).

## 4.3 Adversarial Attacks

We evaluate robustness against four standard state-adversarial attacks, summarized in Table 1. Each attack is applied at every control step during rollouts.

## 4.4 Pruning Strategies

We compare five pruning strategies applied to both actor and critic networks: Random (uniform weight removal under ERK allocation), Magnitude (removing the smallest weights), Magnitude–STE (magnitude pruning with straight-through estimator updates), Saliency (based on first-order Taylor scores), and a dense No-Pruning baseline. All pruning methods (except the baseline) follow a cubic sparsity schedule after a $25\%$ burn-in.

**Micro-pruning.** We call a schedule that increases sparsity through many small, frequent mask updates *micro-pruning*, as opposed to applying a single large pruning step. The global target sparsity still follows a cubic schedule, but the mask is adjusted incrementally so that the model is pruned in fine-grained steps rather than all at once.

**Why gradual steps help.** The three–term bound in Sec. 3 (Theorem 3) shows that pruning introduces a performance loss proportional to $\mathcal{L}_{\mathrm{par}}(\theta, \theta')\,\|\Delta\theta\|$, where $\mathcal{L}_{\mathrm{par}}$ is a path–averaged sensitivity measuring how much Jacobians vary along the pruning trajectory. Large pruning steps can push parameters through regions where sensitivities change sharply, making $\mathcal{L}_{\mathrm{par}}$ large. By contrast, small incremental steps keep consecutive parameters close, so the Jacobian varies smoothly and the integrand in $\mathcal{L}_{\mathrm{par}}$ stays stable. Thus micro-pruning tends to accumulate performance cost more gently, while still benefiting from the monotone decrease in the Lipschitz bound $\tilde{L}_\theta$ that controls robustness.

**Sweet-spot definition.** To quantify the joint effect of pruning on standard performance and robustness, we define the *sweet spot* of each method as the pruning level at which the average of normalized clean and normalized robust performance is maximized. This captures the pruning regime

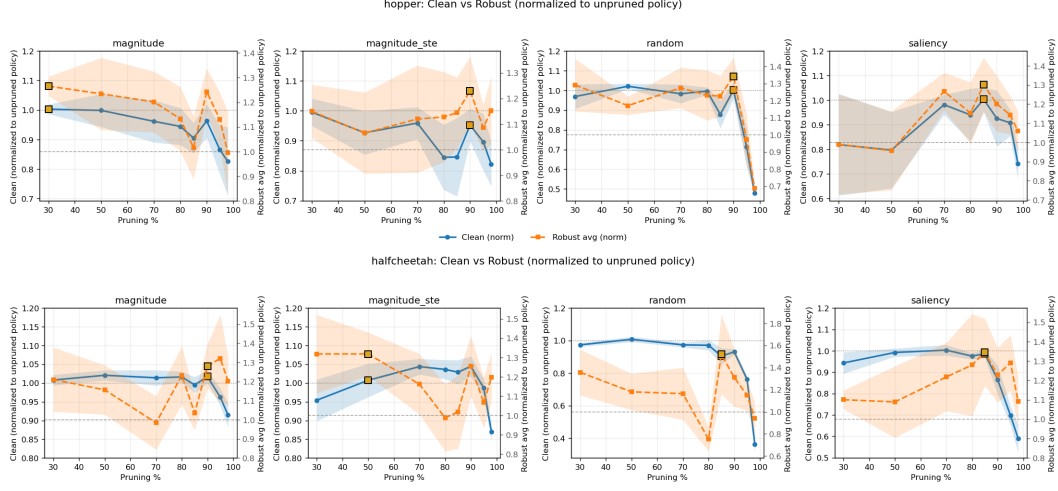

Figure 2: **Clean vs. robust frontiers under pruning.** (Top) `hopper`: normalized clean and robust returns as pruning increases, across strategies. (Bottom) `halfcheetah`: analogous trends with higher pruning tolerance. All curves are normalized to the unpruned SA-trained policy; *shaded regions denote ± one standard error across seeds.*

where robustness improvements are realized without disproportionate loss in clean-task return, and is reported consistently across environments and methods.

## 4.5 EMPIRICAL ANALYSIS

All reported results in this section are *normalized against the unpruned SA-trained network*, which serves as our dense baseline and are averaged over 5 seeds. This ensures pruning is always evaluated relative to the strongest non-sparse policy rather than a weaker vanilla PPO baseline.

Our theory establishes that pruning alone monotonically improves certified robustness (Theorem 1). During training, however, gradient updates can enlarge weight norms and hence $\tilde{L}_\theta$, so robustness is not globally monotone (cf. Training Remark, Sec. 3). Nevertheless, each pruning step strictly decreases $\tilde{L}_\theta$ relative to the current parameters, acting as a monotone regularizer that counteracts the natural growth of weight norms during training. From this perspective, one should not expect perfectly monotone empirical curves, but rather robustness that tends to increase steadily with pruning, punctuated by fluctuations from training noise. We now test this prediction across `hopper`, `halfcheetah`, and `walker2d`, gradually building a picture of how pruning reshapes the performance–robustness landscape.

**Clean vs. robust frontiers.** We begin by examining the overall trade-off between clean-task performance and robustness. Figure 2 shows these frontiers for `hopper` and `halfcheetah`. In `hopper`, robustness climbs until around 40% pruning before clean-task degradation takes over. `halfcheetah` is strikingly more tolerant, maintaining clean-task performance up to ∼70% pruning. These patterns reflect the three-term decomposition in Theorem 3: pruning reduces the Lipschitz gap term, but excessive sparsity eventually drives large parameter displacements that erode performance.

By contrast, `walker2d` is far less forgiving: robustness initially rises but collapses past 50%. These differences align with environment dynamics: `halfcheetah`'s smoother transitions allow redundancy, while `walker2d`'s instability amplifies sensitivity. Appendix C provides the full set of curves, confirming the reproducibility of these trends.

The other environments follow the same pattern but with different tolerances. `halfcheetah` is strikingly robust, maintaining clean-task performance up to ∼70% pruning, whereas `walker2d` is far less forgiving: robustness initially rises but collapses past 50%. These differences align with en-

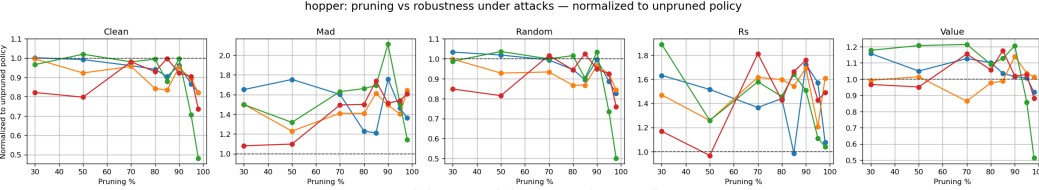

Figure 3: **`hopper` under attack.** Robustness gains are strongest against MAD and RS adversaries, consistent with pruning's global Lipschitz guarantee. Improvements are smaller and less consistent against targeted Value-guided attacks. Appendix C shows analogous plots for `halfcheetah` and `walker2d`.

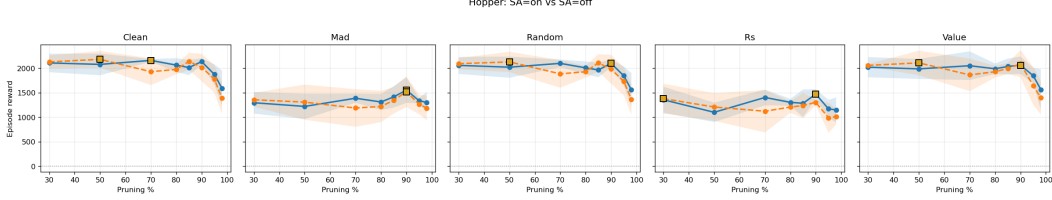

Figure 4: **Pruning vs. adversarial training (`hopper`).** Pruning yields robustness gains in both regimes. SA-regularization sometimes provides additional improvements (notably under RS and Random), but the effect is uneven across attacks.

vironment dynamics: `halfcheetah`'s smoother transitions allow redundancy, while `walker2d`'s instability amplifies sensitivity. Appendix C provides the full set of curves, including seed variability, which confirm the reproducibility of these sweet spots.

The pruning method also matters. Magnitude pruning consistently yields the most stable frontiers, as expected from Theorem 3 since it directly controls $\|\Delta\theta\|$. Saliency pruning looks competitive in `hopper` but breaks down in more complex environments, where instantaneous gradient saliency is a poor proxy for long-horizon contributions. Magnitude–STE introduces noise by pruning sensitive layers too aggressively, and random pruning is unsurprisingly the least reliable: it occasionally boosts robustness but often destroys clean-task returns.

**Attack-specific robustness.** To understand robustness more finely, we next examine performance under different adversaries. Figure 3 shows `hopper` curves under four state-adversarial attacks (Random, Value-guided, MAD, RS). Here a clearer picture emerges: pruning offers the strongest gains against broad-spectrum adversaries (MAD and RS), boosting robust returns by several hundred reward points in the 40–60% sparsity range. By contrast, targeted Value-guided attacks are less affected, producing noisier or weaker gains. This contrast reflects the gap between global and local robustness: pruning reduces the global Lipschitz constant (as guaranteed by Theorem **??**), but does not eliminate vulnerabilities to specific input patterns. In other words, pruning hardens the policy against generic perturbations, but some adversary-specific weaknesses remain. `halfcheetah` and `walker2d` exhibit the same qualitative trends (Appendix C), though the precise sweet spot again depends on environment dynamics.

**Effect of adversarial training.** A natural question is whether pruning simply mimics the effect of adversarial (SA) training. Figure 4 compares `hopper` returns with and without SA regularization. We find that pruning consistently improves robustness in both settings, confirming that it acts as an independent structural bias. The incremental effect of SA training under pruning is modest and attack–dependent (e.g., clearer under RS and Random, negligible under Clean and Value). This suggests that pruning and SA are not interchangeable, but their combination does not always yield additive gains.

**Micropruning ablation.** When pruning is interleaved with training, we observe that robustness gains and clean-task performance often evolve in parallel (Fig. 5) Micro-pruning schedules, which update the pruning mask in small increments, allow the network to adjust gradually: each

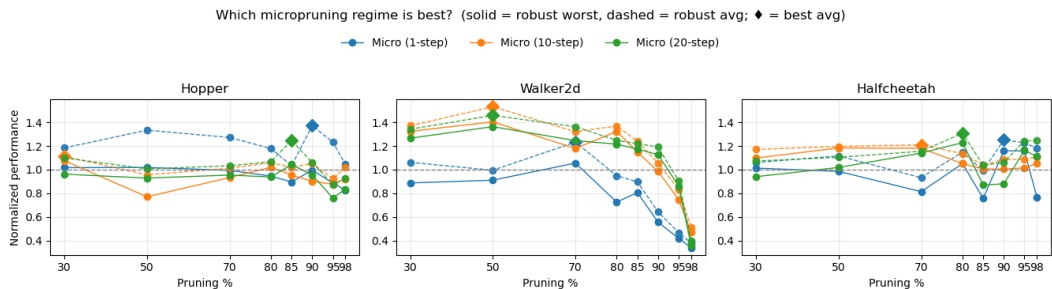

Figure 5: **Micropruning ablation.** Updating masks in small, periodic increments (10–20 steps) leads to smoother curves and more reliable sweet spots than pruning every step.

incremental step reduces the Lipschitz bound controlling robustness, while ongoing weight updates help offset the associated parameter change. As a result, performance curves remain smoother and robustness improvements are more stable, with sweet spots emerging at intermediate sparsity levels. Figure 5 illustrates this effect across pruning intervals. Applying mask updates every 10–20 steps yields the most stable curves, while pruning every single step introduces more variability due to interaction with gradient noise. The same qualitative pattern holds across environments, with `hopper` benefitting most clearly from 20-step pruning, while `halfcheetah` and `walker2d` stabilize at 10–15 steps.

**Sweet spot quantification.** Finally, Appendix Table 2 quantifies the sweet-spot sparsities across environments and pruning methods. `hopper` peaks around 30–50%, `halfcheetah` around 50–70%, and `walker2d` around 30–50%. Notably, magnitude pruning consistently finds these ranges, while random pruning is far more variable. Across tasks, pruning improves normalized robust performance by $1.1\times$–$1.6\times$ relative to baseline, showing that the benefits are substantial and reproducible. Appendix Table 3 further reports per-seed worst-case values, confirming that these gains are not driven by lucky seeds but persist across training runs.

## 5 CONCLUSION AND FUTURE WORK

We studied *element–wise* pruning in reinforcement learning and showed, both theoretically and empirically, that it acts as a monotone regularizer; each pruning step reduces a Lipschitz surrogate of robustness, while performance loss is captured by a three–term regret bound. Across continuous–control benchmarks, we consistently observe reproducible "sweet spots" where robustness gains outweigh clean-task degradation, under both standard and adversarial training. To the best of our knowledge, this is the first work to certify that pruning in RL can never reduce robustness, positioning it not only as a compression tool but as a structural intervention shaping the performance–robustness trade-off. We also acknowledge that large language models (LLMs) were used to assist with LaTeX formatting and grammar checking.

**Limitations and Future Work.** Our study focuses on continuous–control benchmarks with MLP policies and element–wise pruning. While this setting offers a clean testbed, it leaves open important questions regarding generality. Extending the theory and experiments to pixel–based environments and richer architectures (e.g., CNNs, RNNs, or transformers) is a natural next step. Similarly, investigating structured pruning methods—such as neuron, channel, or layer pruning—could provide more practical compression gains and richer robustness–performance trade-offs. Another promising direction is to integrate pruning more tightly with the training process. For instance, jointly optimizing pruning with adversarial or robust training may yield complementary benefits, while data–dependent sensitivity estimates could enable sharper and more adaptive pruning schedules. Beyond the static adversarial models considered here, it is also important to evaluate pruning under richer and more realistic threat models, including adaptive, temporally correlated, or non–stationary attacks. Finally, while we have established pruning as a robustness–preserving intervention, its broader implications for policy generalization, exploration, and sample efficiency remain underexplored. Addressing these questions would help clarify when and how pruning can serve not only as a compression tool but as a principled means of shaping learning dynamics in reinforcement learning.

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

# A    APPENDIX

## A    ALGORITHM

---

**Algorithm 1** PPO with Pruning and Optional SA Regularization

---

**Require:** Initial policy parameters $\theta$; Pruning masks $\{M_\ell\}$ for each layer (initialized as all ones, i.e. no pruning applied); Adversarial budget $B(s)$ (state perturbation set); Regularization weight $\kappa \geq 0$; Total update horizon $T$; Burn-in fraction $\beta \in [0, 1]$; Choice of pruning rule.
 1: Initialize update counter $t = 0$
 2: **for** each update iteration **do**
 3:       Collect trajectories using adversarial states $\hat{s} \in B(s)$
 4:       Compute advantages $\hat{A}$ and returns $\hat{R}$
 5:       **for** each minibatch $\mathcal{B}$ **do**
 6:           $t \leftarrow t + 1$
 7:           Compute PPO loss with SA regularisation $\mathcal{L} \leftarrow \mathcal{L}_{\text{PPO}} + \kappa\, \mathcal{R}_{\text{SA}}(\theta; \mathcal{B})$
 8:           Update network parameters with gradient descent
 9:           **if** $t/T > \beta$ (network burn-in) **then**
10:               Update pruning masks $\{M_\ell\}$ according to chosen rule
11:               Apply masks to network parameters                                    ▷ pruning step
12:           **end if**
13:       **end for**
14: **end for**

---

## B    PROOFS

### B.1    PROOF OF THEOREM 1

**Proof.**  For any policy $\pi$ and its optimal adversary $\nu^*(\pi)$ in the state-adversarial MDP (SA-MDP), it holds from Zhang et al. (2020) that

$$\max_s \left\{ V^\pi(s) - \tilde{V}^{\pi \circ \nu^*(\pi)}(s) \right\} \;\leq\; \alpha \max_s \max_{\hat{s} \in B(s)} D_{\text{TV}}\big(\pi(\cdot|s), \pi(\cdot|\hat{s})\big), \tag{1}$$

where $B(s) = \{\, \hat{s} : \|\hat{s} - s\|_2 \leq \varepsilon \,\}$ is the $\ell_2$ perturbation ball, and $D_{\text{TV}}$ denotes total variation distance.

**Network Lipschitz bound.**    Let the policy network be an $L$-layer feedforward model with parameters $\theta = \{W_1, \ldots, W_L\}$, biases $\{b_\ell\}$, and $L_{\sigma_\ell}$–Lipschitz activations $\sigma_\ell$. Explicitly, the network map $f_\theta : \mathcal{S} \to \mathbb{R}^d$ is the function composition

$$f_\theta(s) \;=\; W_L\, \sigma_{L-1}\Big(W_{L-1}\, \sigma_{L-2}\big(\cdots \sigma_1(W_1 s + b_1) + b_{L-1}\big)\Big) + b_L,$$

where $\sigma_\ell$ is applied elementwise. Biases do not affect Lipschitz constants.

The Lipschitz constant of a linear map $x \mapsto W_\ell x$ is its operator (spectral) norm $\|W_\ell\|_2$, since

$$\|W_\ell x - W_\ell y\|_2 = \|W_\ell(x - y)\|_2 \leq \|W_\ell\|_2 \|x - y\|_2.$$

Thus the Lipschitz constant of $f_\theta$ is bounded by

$$\text{Lip}(f_\theta) \;\leq\; \left( \prod_{\ell=1}^{L-1} L_{\sigma_\ell} \right) \prod_{\ell=1}^{L} \|W_\ell\|_2.$$

Since computing or constraining $\|W_\ell\|_2$ can be difficult, we introduce monotone surrogates. For any matrix $A$,

$$\|A\|_2 \;\leq\; \|A\|_F, \qquad \|A\|_2 \;\leq\; \sqrt{\|A\|_1 \|A\|_\infty}.$$

These upper bounds are monotone in the entries of $A$, hence suitable for analyzing pruning. Therefore,

$$\text{Lip}(f_\theta) \;\leq\; \left(\prod_{\ell=1}^{L-1} L_{\sigma_\ell}\right) \prod_{\ell=1}^{L} \min\Big\{\, \|W_\ell\|_F, \; \sqrt{\|W_\ell\|_1 \|W_\ell\|_\infty}\,\Big\} \;=:\; \tilde{L}_\theta.$$

Thus for any $s, \hat{s} \in B(s)$,

$$\|f_\theta(\hat{s}) - f_\theta(s)\|_2 \;\leq\; \tilde{L}_\theta\, \varepsilon.$$

**Gaussian policies.** For $\pi_\theta(a|s) = \mathcal{N}(\mu_\theta(s), \Sigma)$ with fixed $\Sigma \succ 0$, the closed-form total variation distance between Gaussians with equal covariance gives

$$D_{\text{TV}}\big(\pi_\theta(\cdot|s), \pi_\theta(\cdot|\hat{s})\big) \;\leq\; \frac{\|\mu_\theta(\hat{s}) - \mu_\theta(s)\|_2}{\sqrt{2\pi\,\lambda_{\min}(\Sigma)}} \;\leq\; \frac{1}{\sqrt{2\pi\,\lambda_{\min}(\Sigma)}}\, \tilde{L}_\theta\, \varepsilon.$$

**Categorical policies.** For $\pi_\theta(\cdot|s) = \text{softmax}(z_\theta(s))$, the log-partition function is $1/4$-smooth, which yields the standard bound

$$D_{\text{TV}}(\pi_\theta(\cdot|s), \pi_\theta(\cdot|\hat{s})) \;\leq\; \tfrac{1}{4}\, \|z_\theta(s) - z_\theta(\hat{s})\|_2 \;\leq\; \tfrac{1}{4}\, \tilde{L}_\theta\, \varepsilon.$$

**Effect of pruning.** Let $\theta'$ be obtained by elementwise pruning, $W'_\ell = \mathcal{M}_\ell \odot W_\ell$ with binary masks $\mathcal{M}_\ell$. Each surrogate norm is monotone under pruning:

$$\|W'_\ell\|_F \leq \|W_\ell\|_F, \qquad \sqrt{\|W'_\ell\|_1 \|W'_\ell\|_\infty} \;\leq\; \sqrt{\|W_\ell\|_1 \|W_\ell\|_\infty}.$$

Hence $\tilde{L}_{\theta'} \leq \tilde{L}_\theta$. Activation Lipschitz constants are unchanged, so the same bounds apply with $\tilde{L}_{\theta'}$, which is no larger.

Therefore, for Gaussian ($c = \frac{1}{\sqrt{2\pi\,\lambda_{\min}(\Sigma)}}$) and categorical ($c = \frac{1}{4}$) policies,

$$\max_s \{V^{\pi_{\theta'}}(s) - \tilde{V}^{\pi_{\theta'} \circ \nu^*(\pi_{\theta'})}(s)\} \;\leq\; \alpha\, c\, \tilde{L}_{\theta'}\, \varepsilon \;\leq\; \alpha\, c\, \tilde{L}_\theta\, \varepsilon.$$

Thus pruning cannot worsen the certified robustness bound. $\qquad\square$

**Tightness of the bound.** The surrogate Lipschitz bound $\tilde{L}_\theta$ from Theorem 1 is guaranteed to be monotone under entrywise pruning, but it can be loose compared to the true sensitivity of the policy. A sharper, distribution–dependent refinement is given in Lemma 1:

$$TV(\pi_\theta(\cdot \mid s), \pi_\theta(\cdot \mid s+\epsilon)) \;\leq\; c\Big(\|J_{g_\theta}(s)\|_{\text{op}}\, \epsilon + \tfrac{1}{2}\beta\, \epsilon^2\Big),$$

where $\beta$ is a curvature constant capturing the local variation of the Jacobian, e.g. an upper bound on the Lipschitz constant of $J_{g_\theta}(s)$. This local bound is (trivially) always no larger than the global bound (exact for ReLU networks, up to an $O(\epsilon^2)$ term otherwise), and often much tighter since typical Jacobians have small operator norm. However, unlike $\tilde{L}_\theta$, it is not guaranteed to decrease monotonically under pruning. Thus, the global bound provides provable monotone improvement, while the local refinement better reflects the true robustness landscape but may vary non-monotonically.

**Lemma 1** (Local robustness bound). *Let $\pi_\theta$ be either a Gaussian policy $\pi_\theta(a \mid s) = \mathcal{N}(\mu_\theta(s), \Sigma)$ with fixed $\Sigma \succ 0$, or a categorical policy $\pi_\theta(\cdot \mid s) = \text{softmax}(z_\theta(s))$. Suppose the network outputs $g_\theta(s)$ (mean $\mu_\theta(s)$ or logits $z_\theta(s)$) are $\beta$-smooth, i.e., $\|J_{g_\theta}(x) - J_{g_\theta}(y)\|_{\text{op}} \leq \beta\|x - y\|_2$ for all $x, y$. Then for any perturbation $\epsilon$ and state $s$,*

$$TV\big(\pi_\theta(\cdot \mid s), \pi_\theta(\cdot \mid s+\epsilon)\big) \;\leq\; c\Big(\|J_{g_\theta}(s)\|_{\text{op}} \|\epsilon\|_2 + \tfrac{1}{2}\, \beta\, \|\epsilon\|_2^2\Big),$$

*where $J_{g_\theta}(s)$ is the Jacobian of $g_\theta$ at $s$ (operator norm induced by $\ell_2$), and $c = \frac{1}{\sqrt{2\pi\,\lambda_{\min}(\Sigma)}}$ for Gaussians and $c = \frac{1}{4}$ for categoricals.*

**Proof.** By first-order Taylor's theorem with integral remainder and the $\beta$-smoothness of $g_\theta$,

$$g_\theta(s+\epsilon) = g_\theta(s) + J_{g_\theta}(s)\,\epsilon + \underbrace{\int_0^1 \big(J_{g_\theta}(s+t\epsilon) - J_{g_\theta}(s)\big)\,\epsilon\, \mathrm{d}t}_{r(\epsilon)},$$

and hence

$$\|r(\epsilon)\|_2 \ \leq \ \int_0^1 \|J_{g_\theta}(s + t\epsilon) - J_{g_\theta}(s)\|_{\mathrm{op}} \|\epsilon\|_2 \, \mathrm{d}t \ \leq \ \int_0^1 \beta \, t \, \|\epsilon\|_2^2 \, \mathrm{d}t \ = \ \tfrac{1}{2}\beta \, \|\epsilon\|_2^2.$$

Therefore,

$$\|g_\theta(s + \epsilon) - g_\theta(s)\|_2 \ \leq \ \|J_{g_\theta}(s)\|_{\mathrm{op}} \|\epsilon\|_2 + \tfrac{1}{2}\beta \, \|\epsilon\|_2^2.$$

For Gaussians with identical covariance $\Sigma \succ 0$, the closed-form total variation bound gives $TV\big(\pi_\theta(\cdot \mid s), \pi_\theta(\cdot \mid s + \epsilon)\big) \leq \|\mu_\theta(s + \epsilon) - \mu_\theta(s)\|_2 / \sqrt{2\pi \, \lambda_{\min}(\Sigma)}$. For categoricals, the softmax log-partition is $1/4$-smooth, yielding $TV\big(\pi_\theta(\cdot \mid s), \pi_\theta(\cdot \mid s + \epsilon)\big) \leq \tfrac{1}{4}\| z_\theta(s + \epsilon) - z_\theta(s)\|_2$. Applying these with $g_\theta$ as the mean or logits respectively establishes the claim. $\qquad\square$

## B.2   LEMMA 2

**Lemma 2** (Expected robustness gap). *For any start-state distribution $\mu$,*

$$\mathbb{E}_{s_0 \sim \mu}[V_{\pi_\theta}(s_0) - \tilde{V}_{\pi_\theta \circ \nu^*}(s_0)] \ \leq \ \alpha \, \mathbb{E}_{s \sim d_\mu^{\pi_\theta}}\big[TV(\pi_\theta(\cdot \mid s), \pi_\theta(\cdot \mid \nu^*(s)))\big] \ \leq \ \mathcal{B}(\theta).$$

**Proof.**   The first inequality is obtained from Zhang et al. (2020) by taking the expectation over the difference in values instead of $max_s$. The constant $\alpha$ (defined in the main paper) collects the reward bound and $\gamma$.

For the second inequality, note that by definition

$$TV(\pi_\theta(\cdot \mid s), \pi_\theta(\cdot \mid \nu^*(s))) \ \leq \ TV_{\max}(s; \theta).$$

Taking expectations over $s \sim d_\mu^{\pi_\theta}$ yields

$$\mathbb{E}_s\Big[TV(\pi_\theta(\cdot \mid s), \pi_\theta(\cdot \mid \nu^*(s)))\Big] \ \leq \ \mathbb{E}_s\big[TV_{\max}(s; \theta)\big] = F(\theta).$$

Multiplying by $\alpha$ gives the claimed bound $\mathcal{B}(\theta) = \alpha F(\theta)$. $\qquad\square$

## B.3   LEMMA 3

**Lemma 3** (Clean/attacked value drop under pruning). *Let $\pi_\theta$ be Gaussian with fixed covariance $\Sigma \succ 0$ or categorical softmax with logits $z_\theta(s)$ from a Lipschitz network. Assume that for each $s$, the map $\phi \mapsto g_\phi(s)$ is differentiable almost everywhere. For any pruned parameters $\theta' = \theta - \Delta\theta$, define*

$$\hat{L}_\phi^{\mathrm{par}} := \Big(\mathbb{E}_{s \sim d_\mu^{\pi_\theta}} \|J_\phi g_\phi(s)\|_{\mathrm{op}}^2\Big)^{1/2}, \qquad \mathcal{L}_{\mathrm{par}}(\theta, \theta') := \int_0^1 \hat{L}_{\theta' + t(\theta - \theta')}^{\mathrm{par}} \, \mathrm{d}t.$$

*With $c = \frac{1}{\sqrt{2\pi \, \lambda_{\min}(\Sigma)}}$, $g_\phi = \mu_\phi$ for Gaussian and $c = \tfrac{1}{4}$, $g_\phi = z_\phi$ for categorical, we have*

$$J(\pi_\theta) - J(\pi_{\theta'}) \ \leq \ \alpha \, c \, \mathcal{L}_{\mathrm{par}}(\theta, \theta') \, \|\Delta\theta\|, \qquad \tilde{J}(\pi_\theta \circ \nu^*) - \tilde{J}(\pi_{\theta'} \circ \nu^*) \ \leq \ \alpha \, c \, \mathcal{L}_{\mathrm{par}}(\theta, \theta') \, \|\Delta\theta\|.$$

*(Here $\|\cdot\|$ on parameters is Euclidean, and $\|\cdot\|_{\mathrm{op}}$ is the operator norm induced by $\ell_2$.)*

**Proof.**   By the SA–MDP value–difference bound (Theorem **??** in TV form) applied to two policies at the same state,

$$J(\pi_\theta) - J(\pi_{\theta'}) \ \leq \ \alpha \, \mathbb{E}_{s \sim d_\mu^{\pi_\theta}}\Big[TV\big(\pi_\theta(\cdot|s), \pi_{\theta'}(\cdot|s)\big)\Big].$$

**Gaussian:** For Gaussians with identical covariance $\Sigma \succ 0$, the closed-form TV bound is

$$TV\big(\pi_\theta(\cdot|s), \pi_{\theta'}(\cdot|s)\big) \ \leq \ \frac{\|\mu_\theta(s) - \mu_{\theta'}(s)\|_2}{\sqrt{2\pi \, \lambda_{\min}(\Sigma)}}.$$

**Categorical:** For $\pi = \mathrm{softmax}(z)$, the log-partition is $1/4$-smooth, yielding

$$TV\big(\pi_\theta(\cdot|s), \pi_{\theta'}(\cdot|s)\big) \ \leq \ \tfrac{1}{4}\|z_\theta(s) - z_{\theta'}(s)\|_2.$$

Thus, in both cases,

$$TV\big(\pi_\theta(\cdot|s), \pi_{\theta'}(\cdot|s)\big) \;\le\; c \, \|g_\theta(s) - g_{\theta'}(s)\|_2.$$

Now let $\phi(t) := \theta' + t(\theta - \theta')$, $t \in [0, 1]$. Since $g_\phi(s)$ is (a.e.) differentiable in $\phi$, the fundamental theorem of calculus along $\phi(t)$ gives

$$g_\theta(s) - g_{\theta'}(s) = \int_0^1 J_{\phi(t)} g_{\phi(t)}(s)\,(\theta - \theta')\,\mathrm{d}t.$$

Taking norms and using the operator norm,

$$\|g_\theta(s) - g_{\theta'}(s)\|_2 \;\le\; \int_0^1 \|J_{\phi(t)} g_{\phi(t)}(s)\|_{\mathrm{op}}\,\mathrm{d}t\,\|\Delta\theta\|.$$

By Tonelli/Fubini to exchange expectation and integral, and Cauchy–Schwarz in $s$,

$$\mathbb{E}_s\|g_\theta(s) - g_{\theta'}(s)\|_2 \;\le\; \int_0^1 \Big(\mathbb{E}_s\|J_{\phi(t)} g_{\phi(t)}(s)\|_{\mathrm{op}}^2\Big)^{1/2}\mathrm{d}t\,\|\Delta\theta\| \;=\; \mathcal{L}_{\mathrm{par}}(\theta, \theta')\,\|\Delta\theta\|.$$

Combining with the TV inequality yields the claimed clean-value bound with factor $\alpha c$. The attacked-value bound is identical with $\tilde{J}$, since Theorem **??** holds for robust values with the same TV control. □

### B.4 PROOF OF THEOREM 3

**Proof.** Decompose

$$\mathrm{Reg}_{\mathrm{atk}}(\theta'; \bar\pi) = J(\bar\pi) - \tilde{J}(\pi_{\theta'} \circ \nu^*) = \big[J(\bar\pi) - J(\pi_\theta)\big] + \big[J(\pi_\theta) - J(\pi_{\theta'})\big] + \big[J(\pi_{\theta'}) - \tilde{J}(\pi_{\theta'} \circ \nu^*)\big].$$

The first term is $\mathrm{Reg}_{\mathrm{clean}}(\theta; \bar\pi)$. The second term is bounded by Lemma 2: $J(\pi_\theta) - J(\pi_{\theta'}) \le \alpha c\,\mathcal{L}_{\mathrm{par}}(\theta, \theta')\,\|\Delta\theta\|$. For the third term, apply Theorem 1 to $\pi_{\theta'}$: $J(\pi_{\theta'}) - \tilde{J}(\pi_{\theta'} \circ \nu^*) \le \alpha c\,\tilde{L}_{\theta'}\,\varepsilon$. Summing the bounds yields the claim. □

## C ADDITIONAL RESULTS

This appendix provides the full set of figures and tables referenced in the main text.

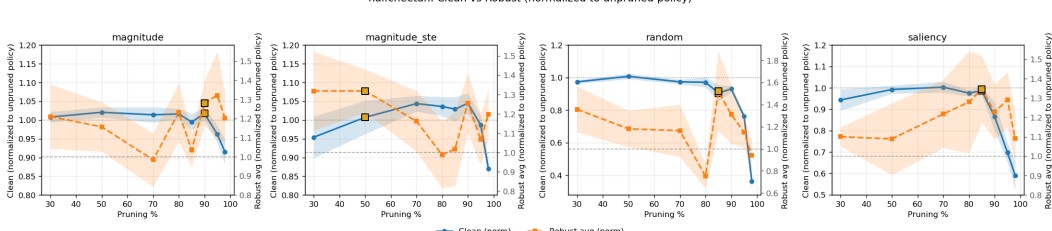

Figure 6: **halfcheetah: Clean vs. Robust frontier.** Normalized clean and robust returns as pruning increases, across pruning strategies.

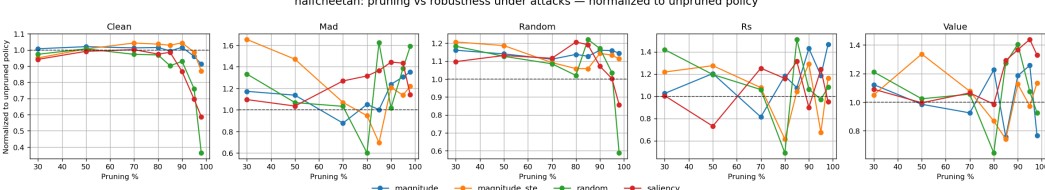

Figure 7: **halfcheetah: Robustness under different adversaries.** Pruning vs. robustness curves across attack types (Clean, MAD, Random, RS, Value).

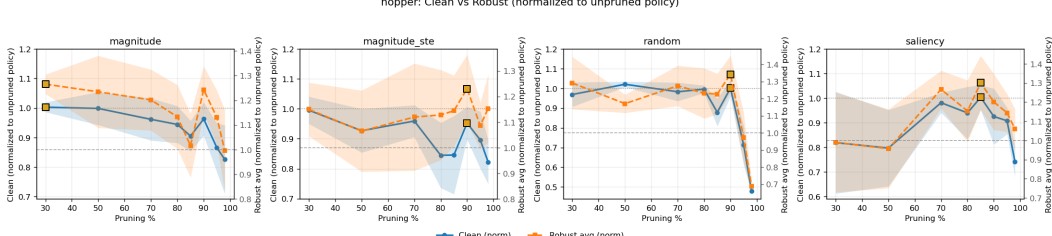

Figure 8: **hopper: Clean vs. Robust frontier.** Normalized clean and robust returns as pruning increases, across pruning strategies.

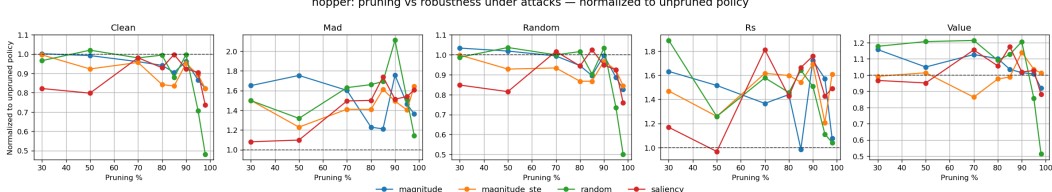

Figure 9: **hopper: Robustness under different adversaries.** Pruning vs. robustness curves across attack types (Clean, MAD, Random, RS, Value).

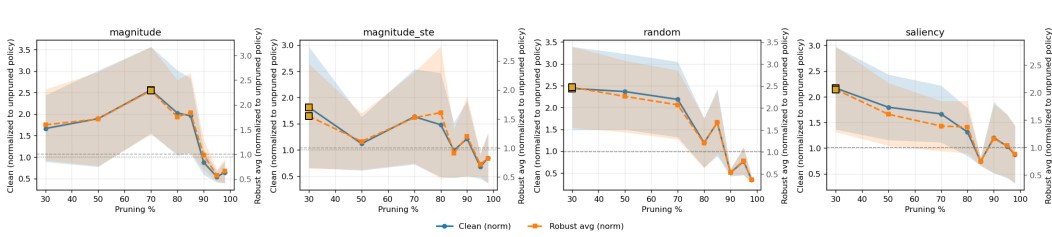

Figure 10: **walker2d: Clean vs. Robust frontier.** Normalized clean and robust returns as pruning increases, across pruning strategies.

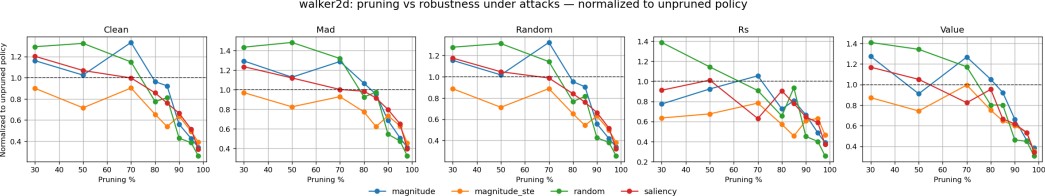

Figure 11: **walker2d: Robustness under different adversaries.** Pruning vs. robustness curves across attack types (Clean, MAD, Random, RS, Value).

| Environment | Method | Sweet-Spot % | Clean | Robust (Worst) | Norm. Robust (Avg) |
|---|---|---|---|---|---|
| hopper | magnitude | 30% | 1.00 | 1.63 | 1.26 |
| | magnitude_ste | 90% | 0.95 | 1.53 | 1.22 |
| | random | 90% | 1.00 | 1.51 | 1.33 |
| | saliency | 85% | 1.00 | 1.66 | 1.28 |
| walker2d | magnitude | 70% | 1.33 | 1.09 | 1.24 |
| | magnitude_ste | 70% | 0.90 | 0.81 | 0.90 |
| | random | 30% | 1.29 | 1.43 | 1.37 |
| | saliency | 30% | 1.20 | 0.94 | 1.13 |
| halfcheetah | magnitude | 90% | 1.02 | 1.24 | 1.23 |
| | magnitude_ste | 50% | 1.01 | 1.47 | 1.28 |
| | random | 85% | 0.90 | 1.62 | 1.35 |
| | saliency | 85% | 0.99 | 1.37 | 1.26 |

Table 2: **Sweet-spot pruning levels.** Across environments, pruning uncovers reproducible sparsity ranges (30–70%) where robustness gains dominate without harming clean returns. Returns are normalized to the unpruned policy performance.

| Environment | Method | Pruning % | Avg. worst-seed abs | Avg. worst-seed norm |
|---|---|---|---|---|
| halfcheetah | magnitude | 98% | 1871.41 | 1.00 |
| | magnitude_ste | 30% | 1908.56 | 1.00 |
| | random | 85% | 1844.73 | 0.91 |
| | saliency | 80% | 2093.46 | 1.11 |
| hopper | magnitude | 30% | 1348.83 | 0.92 |
| | magnitude_ste | 90% | 1227.63 | 0.83 |
| | random | 30% | 1528.69 | 1.06 |
| | saliency | 85% | 1425.76 | 0.96 |
| walker2d | magnitude | 30% | 1123.91 | 0.53 |
| | magnitude_ste | 70% | 446.48 | 0.22 |
| | random | 30% | 2259.16 | 1.11 |
| | saliency | 30% | 1403.68 | 0.68 |

Table 3: For each environment and pruning method (SA=on), we summarize robustness by averaging, over the non-clean attacks, the *worst-seed* mean reward *evaluated at the attack-specific sweet spot*, where the sweet spot is defined as the sparsity that maximizes the *average* reward across seeds within the method. We report a single representative pruning percentage per method as the *mode* of the attack-wise sweet spots (ties favor smaller %). Absolute scores and values normalized to the no-prune (0.30) baseline for each attack are shown.

## D EXPERIMENTAL DETAILS AND CONFIGURATION

**Hyperparameters.** The hyperparameters for PPO are presented in Tables 4

Table 4: Key PPO and attack-specific hyperparameters for adversarial training in `MuJoCo`.

| Hyperparameter | Value |
|---|---|
| Total timesteps | 50M |
| Learning rate | 3e-4 |
| Batch size (envs × steps) | 2048 × 10 |
| Update epochs | 4 |
| Minibatches per update | 32 |
| $\gamma$ (discount factor) | 0.99 |
| GAE $\lambda$ | 0.95 |
| Clipping $\epsilon$ | 0.2 |
| Entropy coefficient | 0.01 |
| Value function coefficient | 0.5 |
| Max gradient norm | 1.0 |
| Adversary hidden size | 256 |
| Similarity penalty $\lambda_{\text{attack}}$ | 10 |
| SA Kappa $\kappa$ | Chosen from $\kappa \in \{0.3, 0.5, 0.7\}$ |

**Experimental compute resources**

All experiments were run as single node jobs across 2 clusters, each comprised of 4 NVIDIA RTX A6000 GPUs - a total of 8 GPUs, each with 48 GB of VRAM.

Upper bounds for compute time are listed below:

- Training victim policies - 109 GPU hours

- Training adversarial policies - 10 GPU hours

- Evaluating adversarial policies against victims - 72 GPU hours

An upper bound for total compute time is $109 + 10 + 72 = 191$ GPU hours, or approximately 8 GPU days.

**Policy network architectures.** We use a unified actor–critic architecture across all `MuJoCo` tasks. The network is implemented in JAX/Flax and consists of two parallel branches for the actor and critic, sharing the same design principle. Each branch is a two-layer MLP with hidden size 256 and either `tanh` or `ReLU` activations (selectable at runtime). Weights are initialized with orthogonal initialization (scaled appropriately), and biases are set to zero.

The actor branch outputs the mean of a diagonal Gaussian distribution over the action space, with fixed covariance $\sigma^2 I$ where $\sigma = 0.1$. Together, these define a Multivariate Normal policy distribution. The critic branch outputs a scalar state-value estimate through its own two-layer MLP with the same hidden size and activation.

This design provides a balanced architecture: compact enough for stable training with pruning, yet expressive enough to capture the dynamics of continuous-control benchmarks.

## E  ADVERSARIAL PERTURBATION VISUALISATIONS

This appendix visualizes the benign observations alongside their corresponding adversarial perturbations for three environments: `Craftax`, `HalfCheetah`, and `Hopper`.

| Natural Observation | Adversarial Perturbation |
|---|---|

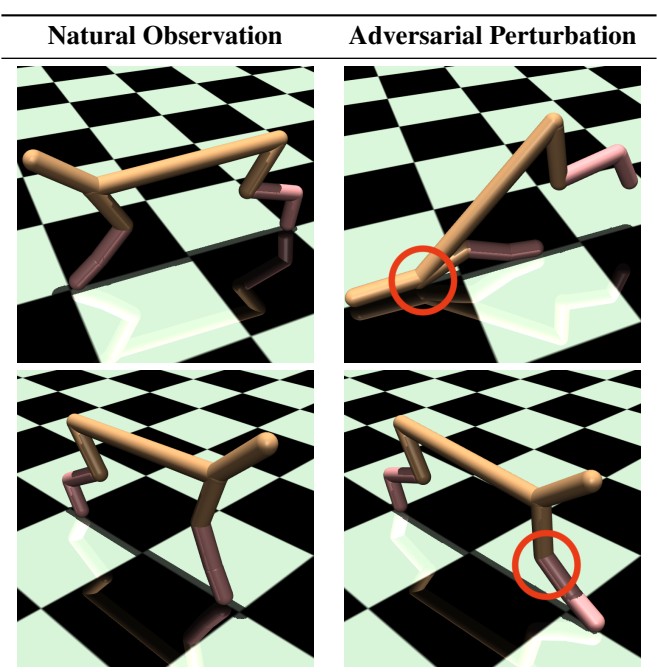

Figure 12: Comparison of benign observations and their corresponding adversarial perturbations in the `HalfCheetah` environment. The first row is a particularly severe perturbation, the kind our adversarial framework is **disincentivized** from producing.

| Natural Observation | Adversarial Perturbation |
|---|---|

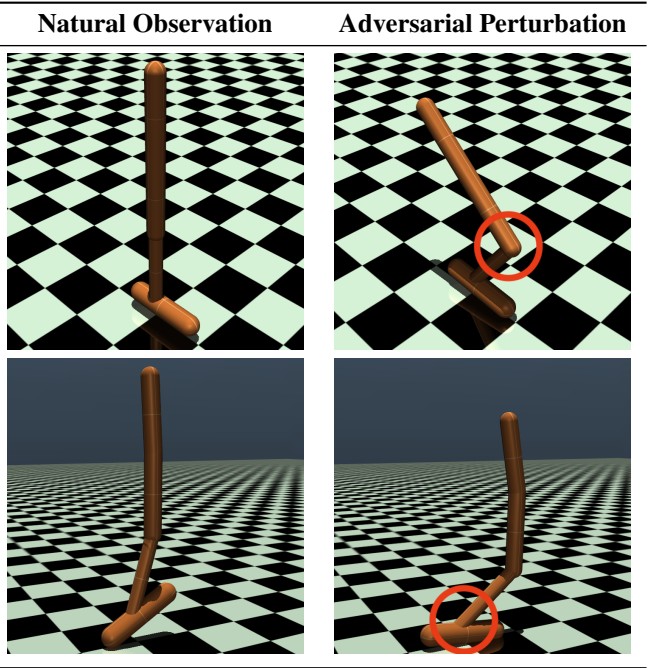

Figure 13: Comparison of benign observations and their corresponding adversarial perturbations in the `Hopper` environment.

## F    CODE AND DATA AVAILABILITY

We include the source code and instructions on how to run our experiments in the supplementary ZIP archive submitted on OpenReview. If the paper is accepted, we will make the code publicly available via GitHub to support transparency and reproducibility.

## G    REPRODUCIBILITY STATEMENT.

We have taken several steps to ensure the reproducibility of our results. Theoretical contributions, including proofs of all main theorems and supporting lemmas, are provided in Appendix B. Full algorithmic details, including pseudocode for PPO with pruning and optional SA regularization, are given in Appendix A. Experimental settings, including hyperparameters, network architectures, compute resources, and pruning schedules, are described in Appendix F. Additional empirical results, including robustness–performance trade-offs across environments, per-seed variability, and micro-pruning ablations, are presented in Appendix C. Visualizations of adversarial perturbations are included in Appendix E. Finally, we provide the full source code and instructions to reproduce all experiments in the anonymous supplementary ZIP archive, which will be made publicly available upon acceptance.

