# OpenReview forum: "Pruning Cannot Hurt Robustness: Certified Trade-offs in Reinforcement Learning"
_ICLR.cc/2026/Conference — ICLR 2026 Conference Withdrawn Submission_

### Official Review · Reviewer_17CR · 2025-10-29

**Soundness:** 3
**Presentation:** 2
**Contribution:** 2
**Rating:** 2
**Confidence:** 5

**Summary:**

The paper addresses adversarial robustness in state-adversarial MDPs (SA-MDPs), and proposes a framework for certified robustness using model pruning. The paper also proposes and analyzes a regret decomposition policy objective, quantifying the tradeoff between robustness and nominal performance. Experimental results justify the theoretical claim that pruning models does not decrease robustness.

**Strengths:**

- The paper provides novel theoretical analyses of an interesting idea, that model pruning has benefits beyond compression, which is likely to be increasingly relevant as the field moves to favor larger and larger models.

**Weaknesses:**

#### Presentation
- Section 4.5, header "Attack-specific Robustness" references a missing theorem.

#### Theorems
- Important implementation details of the method are unclear. Specifically, Theorem 1 references elementwise pruning, but does not define it. What is the condition for pruning a weight, and do these conditions affect Theorem 1?
- $\pi_{\theta} \circ \nu$ is defined implicitly as "the policy under the optimal adversary" (via $\tilde{V}^{\pi_{\theta}\circ\nu}$). It is unclear if this policy is different from $\pi_\theta$. Is the former trained with $\nu$ present in the SA-MDP?
- In Theorem 2, the definition of $Reg_{atk}$ does not make sense. What is the justification for comparing $J(\cdot)$, the unperturbed value of an optimal policy, to $\tilde{J}(\cdot)$, the robust value of the training policy? $Reg_{clean}$ and the combined term are more understandable, in that each keeps either the policy or the function constant.

#### Related Work
- The paper fails to address or compare to any related methods. Please view the following papers, which operate with SA-MDPs: [1,2,3,4,5,6,7]
- Particularly, [4,6,7] deal with regret notions while [1] uses regularization terms quite similar to those used in the paper.

#### Evaluation
- The evaluations are missing comparisons to baseline methods mentioned above.
- Figure 4 shows an ablation on the effect of the SA-regularization versus pruning+SA-regularization. It would be helpful to see the effect of removing pruning as well. Additionally, the fact that the SA-regularizer is stated to be borrowed directly from prior work, and the apparently minimal effect it has on the outcome, raises the question as to why it is used at all.


[1] Tuomas P. Oikarinen, Wang Zhang, Alexandre Megretski, Luca Daniel, Tsui-Wei Weng. Robust Deep Reinforcement Learning through Adversarial Loss. NeurIPS 2021: 26156-26167

[2] Huan Zhang, Hongge Chen, Duane S. Boning, Cho-Jui Hsieh. Robust Reinforcement Learning on State Observations with Learned Optimal Adversary. ICLR 2021

[3] Yongyuan Liang, Yanchao Sun, Ruijie Zheng, Furong Huang. Efficient Adversarial Training without Attacking: Worst-Case-Aware Robust Reinforcement Learning. NeurIPS 2022

[4] Roman Belaire, Pradeep Varakantham, Thanh Hong Nguyen, David Lo. Regret-based Defense in Adversarial Reinforcement Learning. AAMAS 2024: 2633-2640

[5] Yanchao Sun, Ruijie Zheng, Yongyuan Liang, Furong Huang. Who Is the Strongest Enemy? Towards Optimal and Efficient Evasion Attacks in Deep RL. ICLR 2022

[6] Xiangyu Liu, Chenghao Deng, Yanchao Sun, Yongyuan Liang, Furong Huang. Beyond Worst-case Attacks: Robust RL with Adaptive Defense via Non-dominated Policies. ICLR 2024

[7] Roman Belaire, Arunesh Sinha, Pradeep Varakantham. On Minimizing Adversarial Counterfactual Error in Adversarial Reinforcement Learning. ICLR 2025

**Questions:**

- It is stated that pruning cannot hurt robustness, or the gap between performances with and without an adversary present. How does pruning affect raw score? If the upper bound score drops but the lower bound score remains the same, the gap closes. However, such a case is hard to justify as useful since the expected outcome is worse.
- What is the motivation for decomposing regret into clean regret and attack regret?
-The "sweet spot" in Figure 2 is defined as the highest average score between clean and robust scores. Can the method be adjusted to accommodate other objectives, like lowest gap or highest minimum?

---

### Official Review · Reviewer_m4GQ · 2025-10-30

**Soundness:** 2
**Presentation:** 3
**Contribution:** 2
**Rating:** 4
**Confidence:** 4

**Summary:**

This paper investigates the effect of network pruning on the adversarial robustness of deep reinforcement learning policies. This work is motivated by findings in supervised learning, where pruning has been shown to enhance model robustness. The authors extend this line of inquiry to RL, focusing specifically on robustness against state observation perturbations. Theoretically, the paper provides an analysis for Lipschitz network policies before and after pruning. Empirically, this study evaluates four different pruning methods on three MuJoCo continuous control tasks.

**Strengths:**

1. Adversarial robust reinforcement learning against state perturbations is an important problem. To my knowledge, this is the first work to investigate the role of network pruning within this specific setting.
2. This paper is generally well-written and logically structured. The theoretical sections, in particular, are presented with commendable clarity, making the analysis accessible.
3. The authors also exhibit rigor and honesty in their theoretical treatment. They forthrightly acknowledge that the global bound derived in Theorem 1 is loose and supplement this by analyzing a tighter, local bound.

**Weaknesses:**

1.	The paper's core claim that "pruning in RL can never reduce robustness" appears to be an overclaim and requires more rigorous qualification. This assertion is weakened by two factors.

  a)	First, the primary theoretical contributions (pruning’s monotonic robustness improvement and the three-term trade-off) depend on the bound derived in Theorem 1. As the authors acknowledge, this bound is loose, which consequently limits the practical significance of the theoretical guarantees.

  b)	Second, the empirical analysis in Section 4.5 clearly indicates that the effect of pruning on robustness is highly dependent on the specific environment. This task-dependency suggests that broad, generalized conclusions are premature and should be framed more cautiously.

2.	A major concern is the strength of the robustness evaluation. The analysis relies on four attack methods (random, value-guided, MAD, and RS), which are considered relatively weak attacks in the current state-adversarial RL literature. To substantiate the paper's claims, a convincing evaluation should include tests against stronger attacks, such as OPTIMAL [1] and PA-AD [2].

3.	Furthermore, the paper's empirical contribution, while promising, can be significantly developed.

  a)	The conclusions drawn in Section 4.5 are often environment-specific. Expanding the experimental suite to more MuJoCo environments would substantially increase the credibility and generalizability of the findings.

  b)	Additionally, a mismatch exists between the theoretical and empirical scope. The theory addresses both Gaussian and categorical policies, yet the experiments are confined to Gaussian policies. Including an empirical study on categorical policies in discrete action spaces would greatly strengthen the paper's soundness.

4.	Finally, the manuscript's clarity could be improved with careful proofreading. I offer a few examples:

  a)	The term $\tilde{V}$ should better be explicitly defined.

  b)	An inconsistency exists regarding the experimental environments. The experiment results list hopper, walker2d, and halfcheetah, but lines 295-296 discuss settings for the "ant" environment.

  c)	In Section 4.4, Pruning Strategies, it would be better to provide the corresponding citations for each method.

  d)	The paragraph spanning lines 375-405 appears to be largely repetitive of earlier content.

  e)	A reference placeholder ("Theorem ??") is present on line 418.

I would be willing to reconsider my overall assessment if the authors substantially enhance the soundness of the empirical study, particularly by incorporating a more rigorous and convincing robustness evaluation.

[1] Robust Reinforcement Learning on State Observations with Learned Optimal Adversary. ICLR 2021

[2] Who is the strongest enemy? Towards optimal and efficient evasion attacks in deep RL. ICLR 2022

**Questions:**

1.	Theorem 2 appears to be a direct corollary of Theorem 1. Could the authors elaborate on what additional information or distinct insights Theorem 2 provides?

2.	The empirical results are presented in a normalized format. To better assess the baseline's absolute performance and the practical impact of pruning, could the authors provide the absolute scores (raw returns) for both the baseline and the pruned models under both clean and attacked conditions?

3.	Regarding Algorithm 1, line 3:

  a)	How exactly are the "adversarial states $\hat{s}$" generated?

  b)	Does the phrase "collect trajectories using adversarial states $\hat{s}$" mean that actions are sampled using the policy $\pi(\cdot \vert \hat{s})$?

  c)	The baseline [1] noted that sampling from $\pi(\cdot \vert s)$ was more effective for training and implemented SA-PPO based on this. What is the justification for using $\pi(\cdot \vert \hat{s})$ in this work?


1.	What is the precise definition of the "Pruning rate"? Does it refer to the fraction of neurons pruned from the total number of neurons, or the fraction pruned from the subset of neurons that met the pruning criterion?

2.	How is the "robustness" value plotted in Figure 2 calculated?

3.	What do the square markers in Figure 4 represent?

[1] Robust deep reinforcement learning against adversarial perturbations on state observations. NeurIPS 2020 spotlight

---

### Official Review · Reviewer_BYaq · 2025-11-01

**Soundness:** 2
**Presentation:** 2
**Contribution:** 3
**Rating:** 2
**Confidence:** 4

**Summary:**

This paper analyzes the robustness of policies under pruning in state-adversarial Markov processes. It shows that pruning always leads to a more robust policy. A regret bound is derived to characterize the trade-off in pruning. Empirical results are presented to show that pruning uncovers reproducible sweet spots.

**Strengths:**

The paper studies an interesting and relevant problem that connects network pruning with the robustness of learned policies.

**Weaknesses:**

1. The presentation in the paper can be improved. The concept of pruning hasn't been formalized in the paper. The main claim of the experiment section is unclear.
2. The claim that "pruning improves robustness" is not well supported in the theoretical analysis. The analysis in Theorem 1 provides limited insight. It argues that: (1) the Lipschitz constant of the policy is bounded by the norm of the neural network; (2) pruning reduces the norm of the network and improves robustness. However, focusing only on the norm of the network overlooks many important factors. A more fine-grained analysis is needed to better substantiate the claim on robustness.

**Questions:**

See weakness.

---

### Official Review · Reviewer_boYq · 2025-11-01

**Soundness:** 2
**Presentation:** 1
**Contribution:** 1
**Rating:** 2
**Confidence:** 4

**Summary:**

This paper focuses on adversarial reinforcement learning and pruning. This paper provides experiments in the MuJoCo environment solely on two tasks, Hopper and Half-Cheetah, and shows that pruning does not make the policy less robust.

**Strengths:**

Please see below.

**Weaknesses:**

It has already been shown that pruning improves neural network robustness [1]. The fact that pruning would also improve robustness where deep neural networks are used as in deep reinforcement learning, is an expected and predictable consequence of the known results of prior work. The claims and contributions of the submission are rather expected and well-known.

Experiments of the paper are only conducted in two tasks in Hopper and Half-Cheetah. The empirical analysis in the paper is not at a level that verifies the claims made in the paper. Were these tasks cherry-picked? Prior work on this, to verify their claim, provides more substantial and comprehensive evidence.

The submission has several typos. The experiment section is inadequate to even understand or interpret the results let alone reproduce them. Instances of this include but are not limited to just using abbreviations without any proper citations for the adversarial attacks. As a reader I do not know what precisely has been done in this section. The paper needs to be self-contained on its own with proper explanations and citations.

As I read the background section I get the sense that a series of non-contribution colliding papers were cited in a hurry without connecting the prior work to the submission’s contributions. For instance there is a training phase attack section in the background. What is this section for? Is the method proposed in the submission tested against training phase attacks? Are there any results in the paper that show pruning improves robustness towards training time attacks?

The background section refers to a series of papers from almost six years ago on vulnerabilities of deep reinforcement learning and some survey papers on LLM security. There is much more recent work on reinforcement learning security and safety which is about the contributions of the submission. The submission completely misses relevant and recent work which is essential to evaluate the contributions of the submission. Even the adversarial training framework that the submission provides comparison to and is built on [3] is about to be six years old and it is shown to have several critical issues with regard to robustness and security; from its vulnerability to black-box adversarial attacks [4] to its natural non-robustness and generalization issues [5].

More critically, even the papers that are cited in the submission do not match the description provided by the submission. For instance, the submission cites this paper [6] with the following description “test-phase transferability attacks”. However, this paper [6] is not about transferability of attacks.

This is either a function of the authors not reading the papers they are citing or they are using a large language model to write. Either of these possibilities is critically concerning.

The contributions of the paper are rather expected and known from existing work. The experimental evaluation is weak, the background section is inadequate and fails to place or explain the submission's contributions, and the writing of the submission is unclear. Given these issues, I do not think this paper is ready to be published


[1] HYDRA: Pruning Adversarially Robust Neural Networks, NeurIPS 2020.

[2] How Sparse Can We Prune A Deep Network: A Fundamental Limit Perspective, NeurIPS 2024.

[3] Robust deep reinforcement learning against adversarial perturbations on state observations, NeurIPS 2020.

[4] Deep Reinforcement Learning Policies Learn Shared Adversarial Features Across MDPs, AAAI 2022.

[5] Adversarial Robust Deep Reinforcement Learning Requires Redefining Robustness, AAAI 2023.

[6] Enhanced adversarial strategically-timed attacks against deep reinforcement learning, ICASSP 2020.

**Questions:**

Please see above.

---

### Note · Authors · 2025-11-14

I have read and agree with the venue's withdrawal policy on behalf of myself and my co-authors.